# Implementation of a colorectal cancer screening intervention in Malaysia (CRC-SIM) in the context of a pandemic: study protocol

Désirée Schliemann [ID],[1] Kogila Ramanathan,[2,3] Nor Saleha Binti Ibrahim Tamin,[4] Ciaran O'Neill,[1] Christopher R Cardwell,[1] Roshidi Ismail,[3] Prathibha Nanu,[5] Ahmad Nizal bin Mohd Ghazali,[6] Frank Kee,[1] Tin Tin Su [ID],[2,3] Michael Donnelly[1]

For numbered affiliations see end of article.

**Correspondence to**
Dr Désirée Schliemann;
D.Schliemann@qub.ac.uk

## ABSTRACT

**Introduction** Colorectal cancer (CRC) is the second most common cancer in Malaysia and cases are often detected late. Improving screening uptake is key in downstaging cancer and improving patient outcomes. The aim of this study is to develop, implement and evaluate an intervention to improve CRC screening uptake in Malaysia in the context of the COVID-19 pandemic. The evaluation will include ascertaining the budgetary impact of implementing and delivering the intervention.

**Methods and analysis** The implementation research logic model guided the development of the study and implementation outcome measures were informed by the 'Reach, Effectiveness, Adoption, Implementation and Maintenance' (RE-AIM) framework. This CRC screening intervention for Malaysia uses home-testing and digital, small media, communication to improve CRC screening uptake. A sample of 780 people aged 50–75 years living in Segamat district, Malaysia, will be selected randomly from the South East Asia Community Observatory (SEACO) database. Participants will receive a screening pack as well as a WhatsApp video of a local doctor to undertake a stool test safely and to send a photo of the test result to a confidential mobile number. SEACO staff will inform participants of their result. Quantitative data about follow-up clinic attendance, subsequent hospital tests and outcomes will be collected. Logistic regression will be used to investigate variables that influence screening completion and we will conduct a budget impact-analysis of the intervention and its implementation. Qualitative data about intervention implementation from the perspective of participants and stakeholders will be analysed thematically.

**Ethics and dissemination** Ethics approval has been granted by Monash University Human Research Ethics Committee (MUHREC ID: 29107) and the Medical Review and Ethics Committee (Reference: 21-02045-07G(2)). Results will be disseminated through publications, conferences and community engagement activities.

**Trial registration number** National Medical Research Register Malaysia: 21-02045-07G(2).

## INTRODUCTION

Colorectal cancer (CRC) is the most common cancer among males in Malaysia and second

## STRENGTHS AND LIMITATIONS OF THIS STUDY

⇒ The South East Asia Community Observatory dataset provides population-level data, a frame for random sampling and an aid to improving the rigour of methods and analysis (eg, checking for systematic bias).

⇒ The focus on measuring implementation as well as conducting a budget impact analysis will provide lessons and insights into scaling up home testing and its wider evaluation in a middle-income country.

⇒ The person-centred, mixed-methods approach will be used to evaluate barriers and facilitators to participating in home testing.

⇒ This is a feasibility study and it does not assess the effectiveness and cost-effectiveness of colorectal cancer home testing.

most common cancer among females and over 70% of patients with CRC are diagnosed at stage III and IV.[1] Late detection requires more intensive treatment, increases healthcare resource utilisation and places additional financial burden on households.[2,3] Screening is key for the early detection and successful treatment of cancer. CRC screening guidelines in Malaysia recommend that primary clinicians should conduct opportunistic screening using the immunochemical feacal occult blood test (iFOBT) in patients who are aged 50–75 years without a family history of CRC and who present asymptomatically.[4] The iFOBT is a stool test that detects blood in the stool, a common sign of CRC. The CRC screening pathway (figure 1) describes the Ministry of Health (MOH) Malaysia guidance and the steps that follow a symptom assessment of patients.[4] Despite the guidance and the provision of free opportunistic screening, CRC screening uptake is <3% among the target population.[5] A number of factors contribute to low CRC screening uptake in

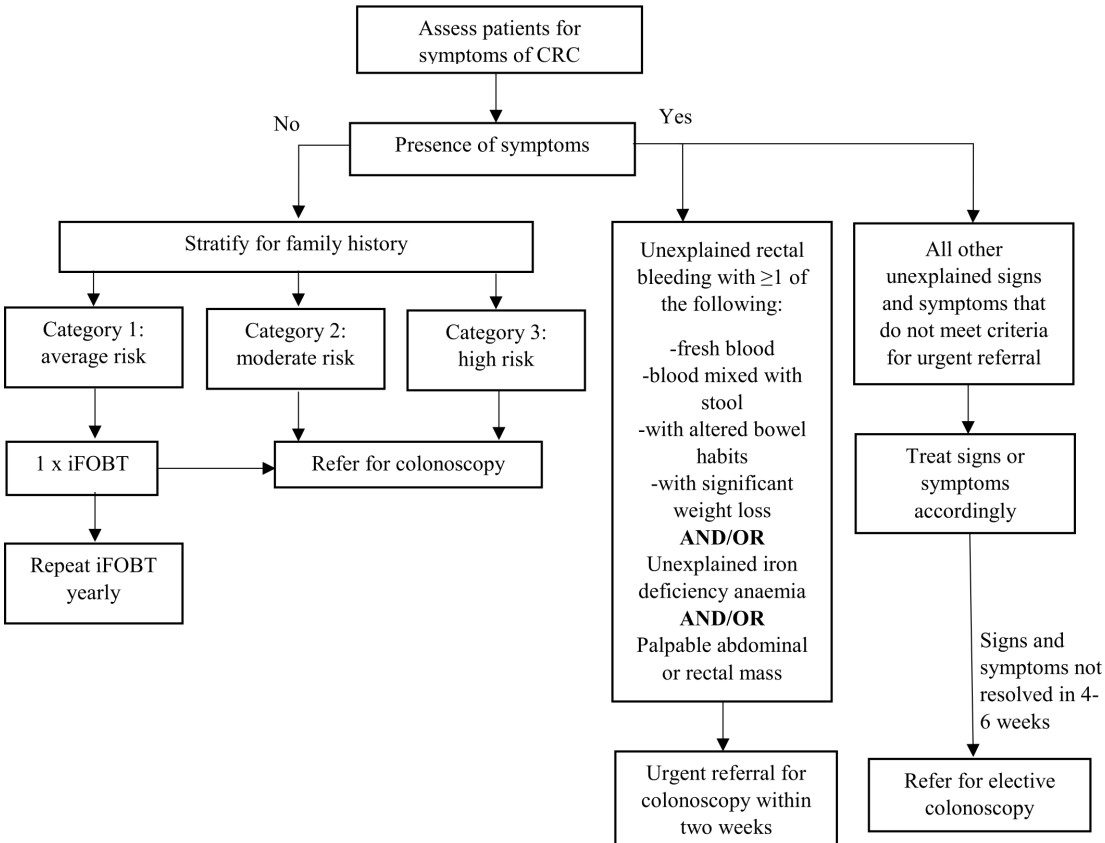

**Figure 1** Current CRC screening pathway, Ministry of Health Malaysia. CRC, colorectal cancer; iFOBT, immunochemical feacal occult blood test.

Malaysia such as low CRC awareness,[6] negative beliefs about cancer, fear of the result, financial concerns and absence of a doctor's recommendation.[7–9] Understanding about CRC tests is lacking[9 10] and willingness to participate in CRC screening is low.[11] The COVID-19 pandemic and the resulting movement control orders (MCOs) in Malaysia, which enforced travel restrictions to limit the spread of the virus, add further barriers to accessing health services and contributing to an increase in CRC cases that are detected late. There is a need for a population-based screening programme that addresses these barriers to detect cancer early and improve survival.[12 13] The healthcare system and context in Malaysia requires systematic research attention in order to improve CRC screening and its uptake. The research findings will be used to inform evidence-based discussions about the implementation of a population-based CRC screening programmes during the pandemic and in the future.

### Aims

Our main aim is to design, develop, implement and evaluate an intervention to improve CRC screening uptake in Malaysia in the context of a pandemic (including budgetary impact of the intervention). The specific objectives are to:

1. Review and synthesise best available evidence and implementation lessons; and assess views and preferences of stakeholder groups.

2. Codesign a culturally sensitive population-based CRC screening uptake intervention for Malaysia and evaluation metrics for the assessment of its implementation.

3. Conduct a study of the implementation of population-based CRC screening uptake in order to test its appropriateness, feasibility and acceptability during the COVID-19 pandemic.

4. Develop a protocol for a pilot evaluation of the implementation model using an experimental analytical design in order to inform a potential nationwide scaled-up programme including planning requirements, resources and costs.

### DESIGN AND METHODS

This section presents information about intervention design, development and implementation, and about the quasi-experimental study plan and outcome measures that will be used to evaluate the CRC screening intervention for Malaysia (CRC-SIM). The Template for Intervention Description and Replication (TIDieR) checklist guided the description of the intervention.[14]

### Patient and public involvement

This study was designed in collaboration with the MOH, academics and South East Asia Community Observatory (SEACO) community engagement committee (CEC) members who are representatives of the local community

in the study district. The involvement of CEC members via group discussions and stakeholder meetings led to the final design of the CRC-SIM intervention.

## Setting

Malaysia has a multiethnic population with communities from mainly Malay (69.9%), Chinese (22.6%) and Indian (6.8%) ethnic backgrounds.[15] The SEACO at Monash University Malaysia is a comprehensive database of over 44 000 people living in Segamat, Malaysia, which is 85% of the population in 5/11 subdistricts and 24% of the total population in Segamat. It captures detailed longitudinal sociodemographic and health-related data about Segamat residents; and in the case of this protocol it provides a research platform for a focused study of screening implementation. Participants will be recruited from two subdistricts, that is, Sungai Segamat (semiurban) and Gemereh (rural). The selection of the subdistricts was made in consultation with local stakeholders. Participants will be trained, remotely, to complete the CRC test in their own home and they will be referred to their local health clinic if the test result is abnormal. Segamat District Hospital has agreed to facilitate up to three colonoscopies per week for this project if required (on top of up to 10 colonoscopies/week conducted currently).

## Study population

The target population will comprise residents from Sungai Segamat and Gemereh who are recorded in the SEACO database and, previously, gave consent to be included in studies. The population of these two subdistricts represents all three ethnicities in Malaysia though Indians are underrepresented (2% from Sungai Segamat/Gemereh vs 9% overall registered with SEACO). Indian Malays living in the two subdistricts will be invited to participate in this study in order to ensure representativeness of the key ethnic groups. Mobile phone ownership among residents aged 50–75 years from the study subdistricts is similar to residents registered with SEACO overall (77% and 78%, respectively). We will include men and women aged 50–75 years who participated in the most recent SEACO health survey (2018) and registered a mobile phone number with SEACO and have access to a smartphone. This age range is in keeping with MoH guidance about CRC opportunistic screening practice in Malaysia. Participants with a history of CRC and/or experiencing CRC symptoms at the time when a researcher calls will be excluded from the study. Data collectors will read a list of common CRC symptoms to participants and ask whether or not they experience any of these symptoms. We will encourage participants who are reporting CRC symptoms to see a healthcare professional as soon as possible.

## Evidence synthesis and intervention design

We conducted an evidence synthesis and local needs assessment that included three investigative steps: (1) a review of the CRC screening programme delivered by the Public Health Agency in Northern Ireland (NI) and CRC screening delivered by the MOH Malaysia; (2) a scoping review of studies of CRC screening in low-income and middle-income countries (LMICs) and (3) focus group discussions (FGDs) and interviews with residents from the target population and key stakeholders in the study area in Malaysia.

In brief, NI introduced a population-based screening programme in 2010. Residents who are registered with a general practitioner and are aged 60–74 years are invited, every 2 years, to participate in bowel cancer/CRC screening. The target population receives a letter, instruction leaflet and the FOBT kit via surface mail and each person is asked to return three stool samples by mail to the Public Health Agency for laboratory testing. Residents who do not return the FOBT kit stool samples within 6 weeks receive a reminder letter. FOBT kit stool sample completion increased from 46% in 2010 to 60% in in 2017.[16] Commonly reported barriers to returning the FOBT kit were lack of awareness about CRC and its prevalence and complexity of the kit.

CRC screening in Malaysia is opportunistic rather than population based. Ideally, patients who are aged 50–75 years will be offered a yearly iFOBT if and when they attend their local health clinic. Patients with CRC who present symptomatically are referred for a colonoscopy (without an iFOBT). A recent study of iFOBT screening by the MoH reported that 127 957 iFOBTs were performed between 2014 and 2018, which was the equivalent of 2.29% of the population being screened and repeat screenings are unusual. About 10% of the iFOBTs were positive (n=11 782/127 957), about half of positive patients (n=6491/11 782) underwent a colonoscopy and about 4% of patients (n=262/6491) were diagnosed with CRC.[5] The most commonly reported barriers in a CRC screening study of 3559 Malaysians were a patient's disgust about stool collection, fear of cancer, poor awareness about the availability of CRC screening and the absence of a doctor's recommendation; access to a colonoscopy service, limited healthcare facilities and trained staff were reported barriers to upscaling of screening; and knowledge of risk factors, perceived susceptibility and a doctor's recommendation positively influenced uptake and completion of an iFOBT.[17]

## Scoping review

Our mapping of the literature about the implementation and evaluation of CRC screening in LMICs[18] suggested that various intervention or programme designs may be successful in terms of achieving a FOBT/iFOBT uptake of ≥65%, which is the recommended target screening uptake according to the European CRC screening guidelines.[19] Successful interventions tended to comprise one or more of the following features: face-to-face interaction with a participant in their home, clinic or public space; opportunistic recruitment of participants in clinics; education session(s) alongside a stool test; a tested and tailored screening protocol; linkage to a healthcare or

cancer registry; and organised and delivered collaboratively by an interagency. Conversely, interventions that were least successful or that achieved less than the lowest acceptable iFOBT/FOBT uptake of 45%,[19] recruited participants through letter or media only or from a workplace or health club; offered a colonoscopy as the primary screening test or delivered an educational session that recommended screening but did not offer iFOBT/FOBT.[18] Findings were inconclusive or mixed for interventions that comprised other features such as a risk assessment questionnaire.

## Qualitative research

We conducted 11 FGDs with 89 members of the local community in Segamat aged ≥50 years in 2019. FGDs were conducted separately with males and females from each ethnic group (Malay, Chinese, Indian and Orang Asli). The thematic analysis suggested that community members had limited awareness about CRC and screening, trusted and followed their doctor's advice and expressed fear about the FOBT/iFOBT result, disgust regarding stool collection and financial concerns regarding the cost of cancer screening and treatment. Some members reported that they lacked access to transport. Participants recognised the importance of early screening for cancer and appeared to use local clinics for treatment and management of their ill-health (eg, diabetes and high blood pressure). The use of the postal service for CRC screening and receiving or sending a stool container was not trusted, and concern was expressed about literacy levels in communities. The recruitment strategy that received most support from participants involved (1) local clinics posting an invitation letter to eligible community members for an FOBT/iFOBT, (2) the collection of a stool container from 'the counter' in each clinic (as a way of avoiding long waiting times), (3) the return of a stool sample to the clinic in-person and (4) receipt of the CRC screening result in writing or over the phone (if negative) and an in-person consultation with a doctor, if positive.

We also conducted 24 individual interviews with key stakeholders (ie, doctors, surgeons, nurses, nongovernmental organisations (NGOs), community health workers, MOH officials and post officers) to elicit their views about organisational readiness, barriers, benefits and enablers for CRC screening in Malaysia. Interviewees expressed concerns about the health system such as the limited availability of CRC data and inconsistent reporting, the relative low priority afforded to noncommunicable diseases (NCDs), particularly cancer, and the level of funding allocated for CRC—these concerns were perceived as contributing to the low CRC screening rates in Malaysia. Furthermore, a number of clinic-level resource constraints were noted such as understaffing, limited availability of iFOBT kits, short appointment times and colonoscopy capacity as well as the lack of salience among doctors about CRC awareness and screening recommendations. Interviewees reported that patient-related issues included a lack of awareness among patients about the importance of early CRC screening and detection, illiteracy and language barriers, access to transport and negative perceptions of iFOBT (disgust about stool collection, fear of cancer) and colonoscopy (fear of discomfort). Interviewees expressed support for collaborative initiatives between NGOs, private clinics and community health volunteers that supplemented clinic-related resource constraints regarding awareness activities, patient navigation and screening.

## Conceptual framework

The implementation of the intervention was informed by our evidence synthesis and implementation research logic model (IRLM)[20] (online supplemental file 1). The IRLM brings together key principles from implementation science and theoretical frameworks (such as the Consolidated Framework for Implementation Research,[21] Expert Recommendations for Implementing Change,[22] RE-AIM,[23] the Theory of Change and the Theory of Action) in the form of a 'map' to navigate the determinants, strategies, mechanisms of action and outcomes of intervention implementation (online supplemental file 1).

## Intervention

We translated and adjusted the findings from the evidence synthesis to inform the design and development of an uptake intervention that will be implemented in the context of the COVID-19 pandemic, and, therefore, will involve only minimal human contact and travel. Although the qualitative interviews (that were completed prior the pandemic) suggested that participants were willing to collect the stool container from their nearest health clinic, this was not possible and had to be adapted due to the imposed travel restrictions during the pandemic. Feedback from community members suggested that it was highly likely that participants would be responsive to recruitment through letter if it came from a trusted organisation, that is, a clinic or SEACO, however, participants also queried the trustworthiness of a postal screening intervention without any upfront notice. Therefore, the screening uptake intervention will combine home-testing with a mix of small media methods to communicate with the target population (figure 2).

The intervention, including recruitment, intervention implementation and follow-up data collection, will take place between August 2021 and March 2022. SEACO will recruit randomly selected, eligible members of the target population over the phone due to the MCO imposed by the COVID-19 pandemic. During the recruitment phone call, participants will be informed about the purpose of the study, the screening intervention and that participating in the iFOBT is free of charge. We will aim to phone about 100 participants each week. This number is based on the capacity of the local hospital to facilitate colonoscopies. A CRC home-screening 'pack' will be delivered by courier to residents who agree to participate and meet the inclusion criteria. Each pack will comprise a personalised letter, an

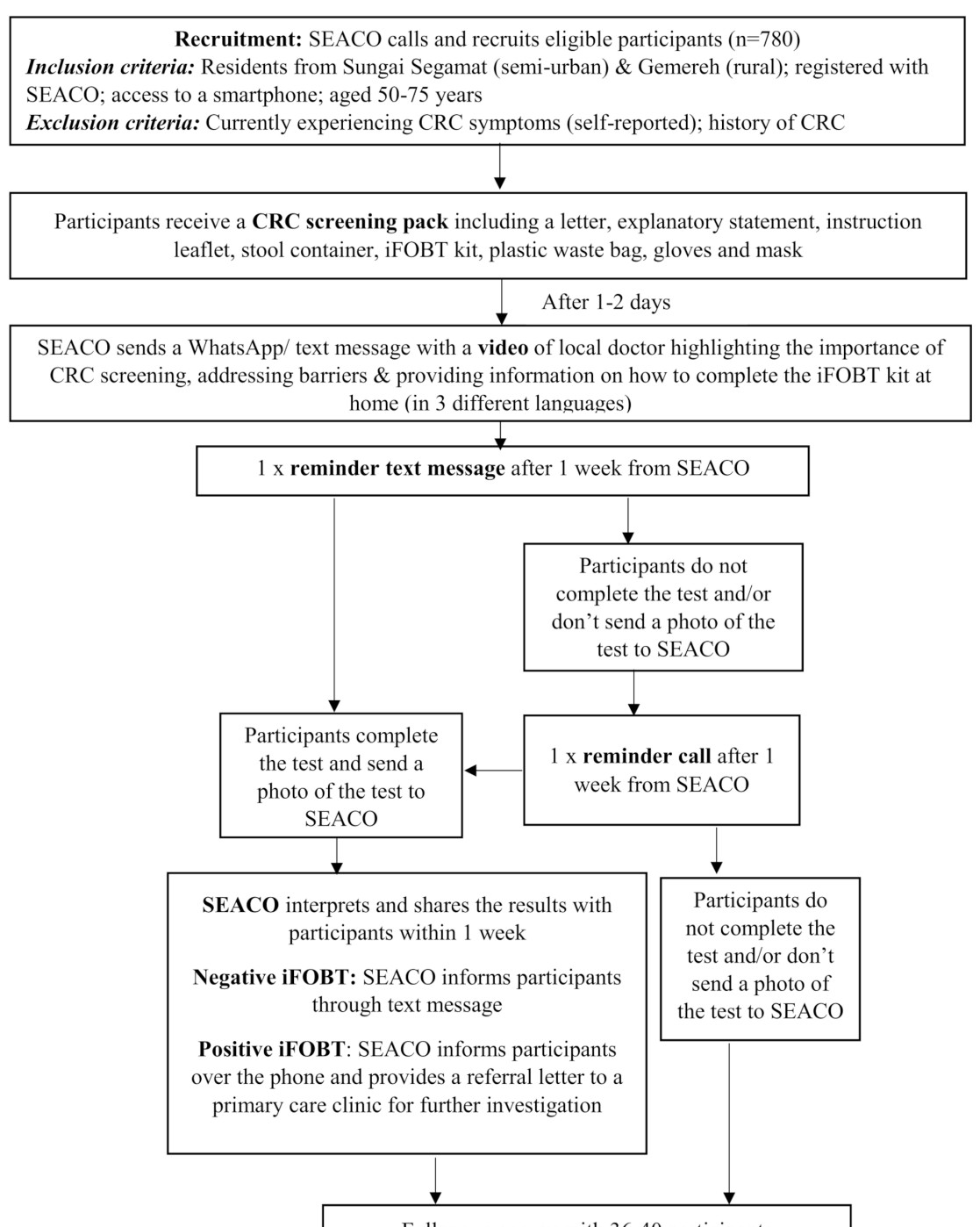

**Figure 2** CRC-SIM intervention flow. CRC, colorectal cancer; CRC-SIM, CRC screening intervention for Malaysia; iFOBT, immunochemical feacal occult blood test; SEACO, South East Asia Community Observatory.

explanatory statement, an iFOBT kit, a stool container, an illustrated leaflet about how to collect a stool sample and use the test kit, a glove, face covering to mask the smell and a plastic waste bag. Participants will receive 1–2 days later, a research-informed video clip through WhatsApp messenger or a text message from SEACO in the preferred language of a participant. The video clip will show a local doctor explaining the importance of the early detection of CRC, how to collect a stool sample and complete the test, as well as addressing common barriers such as disgust and fear of the test results. The video will also explain how to share the result with SEACO and the importance of a colonoscopy in case of a positive iFOBT result. This intervention will address inhibiting factors (ie, lack of awareness, emotional barriers, literacy and language barriers, short appointment times with doctors, lack of transport to clinics) and use facilitating factors (trust in a doctor, health education, interagency collaboration) for CRC screening. Participants will be asked to send a photograph of the test result to a SEACO

confidential telephone number. A research assistant will send a text message reminder after 1 week to all participants to complete the test and share the result. Participants who did not share their iFOBT result after 2 weeks will be called and reminded by the research assistant. Two trained medical professionals who are members of the research team will interpret the results independently and compare them for accuracy. On agreement, the results will be shared with the participants through text message if negative and telephone call. Participants with invalid or inconclusive results will be asked to perform a second iFOBT-test at home. In the case of a second invalid test result, participants will be asked to complete an iFOBT at their local health clinic. Clinicians will not be reimbursed for follow-up patient appointments and colonoscopies but will be facilitated within existing resources. Participants with a positive result will receive a phone call and text message containing a photograph of a referral letter. The referral letter will suggest to participants to visit their doctor at their nearest public health clinic who will refer the participants to the hospital colonoscopy service. Once iFOBT positive participants have visited the health clinic, they will follow the treatment pathway as recommended by the MOH. Participants will be reminded to screen annually for CRC after participating in this study, that is, during the recruitment telephone call and in the text message sent to iFOBT negative participants.

## Training and piloting

All materials (eg, letter, leaflet, video, text message reminder) will be codesigned together with local clinicians and community members to ensure acceptability and cultural appropriateness. Five SEACO staff from various professional backgrounds will be recruited and trained to conduct recruitment and follow-up calls and record iFOBT-related research data. Recruited staff will receive a 2-day training in data collection and recruitment (including role play) by SEACO that will be concluded with a test that staff has to pass (80% passing mark). One experienced SEACO supervisor with a BSc in Social Sciences will supervise the recruitment and data collection. Two clinically trained research staff will interpret the iFOBT results. SEACO staff need to demonstrate excellent communication, computer and record-keeping skills to be considered for this role.

Intervention delivery and study procedures will be piloted with 20 participants from the two participating sub-districts and the results will be used to fine-tune the intervention flow and data recording procedures before the implementation of the main study.

## Evaluation
### Outcome indicators

Our primary objective is to describe and evaluate the implementation of the CRC-SIM. The RE-AIM framework, that is, reach, effectiveness, adoption, implementation (including a budget impact analysis) and maintenance,[23] together with measures of acceptability, appropriateness

and feasibility[24] provide the key evaluation outcomes. We will gather a range of quantitative and qualitative data about these indicators and conduct a budgetary impact analysis (table 1).

## Methods of assessment
### Participant information

Information about gender, age, ethnicity, income, education, occupation, current NCDs (eg, diabetes and hypertension), body mass index, smoking, cancer history and ownership of motorised vehicle will be extracted from the most recent health survey (2018) recorded in the SEACO database in order to present a profile of study participants (including comparisons of residents who agreed or refused to participate). Participants who refuse to participate will be reported as 'non-participants' and participants who wish to withdraw from the study after they receive a test-kit will be reported as 'did not complete the iFOBT' including the reasons given for withdrawal.

### Interviews

Qualitative interviews will contribute to the assessment of intervention implementation, particularly its acceptability to, and accessibility by, the local community. Following the implementation of the uptake intervention, interviews will be conducted with a subsample of study participants as well as all involved healthcare staff from Sungai Segamat health clinic and CRC surgeons from Segamat hospital. A purposively selected subsample of study participants will be recruited over the phone to gain insights about each point of the intervention strategy as well as participants who completed all and who completed no part of the intervention. Study participants will be selected based on their ethnicity, gender and completion of study components (ie, non-participants, participants who did/did not complete the stool sample, iFOBT positive/negative cases and participants who did/did not attend their colonoscopy appointment). Healthcare staff who were involved in the CRC-SIM will be recruited through email (6–8 participants). A combination of quantitative and qualitative interview questions have been informed by the NI Bowel Cancer Screening evaluation,[25] by the RE-AIM framework[23] and by an adapted version of the validated the acceptability e-scale questionnaire.[26] Trained, bilingual research assistants will interview 36–40 participants or until data saturation occurs and interviews stop generating new insights about each point of the intervention strategy. Interviews with participants who completed the iFOBT and received a negative result will take place 1–2 weeks after they received their result, therefore, some participants will be interviewed while data collection is still ongoing. The quantitative and qualitative questions will be piloted with a small number of participants (n~10) from the pilot study.

### Clinical outcomes

Two trained medical professionals in the research team will interpret separately, the same first series of photographs

**Table 1** Outcome measures

| Outcome measure | Details | Mode of assessment |
|---|---|---|
| Reach | ► n/% of participants in various characteristic-defined groups (eg, ethnic groups) who agreed to participate, received/read the letter/leaflet and watched the video<br>► n/% who participated at each point of intervention process (see below) | ► Recruitment and follow-up call by SEACO staff<br>► Interview |
| Effectiveness | **Service-related outcomes:**<br>► n/% of test photos returned to SEACO<br>► n of days between recruitment, delivery of test kit, sharing of test kit with SEACO (photo), sharing of test results with participants<br>► n of negative results shared with participant through text message<br>► n of positive results shared with participant through telephone call<br>► n of weeks between positive test, doctor's appointment and colonoscopy appointment<br>**Patient-related outcomes:**<br>► Date of photo sent<br>► n/% of positive/negative iFOBTs<br>► n/% of colonoscopies (out of all FOBT positive participants)<br>► n/% of participants that follow-up a positive result with their doctor<br>► Colonoscopy result<br>► Costs (see below) | ► Spreadsheet to capture return of photographs<br>► Data recorded by SEACO staff and healthcare staff |
| Adoption | ► Description of training (SEACO staff)<br>► n of SEACO staff trained and staff background/experience<br>► n of trained SEACO staff who complete training and stay for the duration of the study<br>► Variation in adoption of intervention uptake between clinics/settings/ areas (doctor's appointments and referral for colonoscopy) | ► Data recorded by research staff |
| Implementation | ► Fidelity of SEACO staff to intervention process and procedures<br>► Participant/patient adherence to intervention procedures (see above)<br>► Cost analysis (see below)<br>► n of reminder text messages<br>► n of reminder calls<br>► Duration of reminder calls /recruitment calls | ► Interview<br>► Phone-related data recorded by SEACO staff |
| Maintenance | ► Willingness of participants and key stakeholders to maintain the intervention<br>► Attrition rate | ► Interview<br>► Spreadsheet to capture return of photos |
| Acceptability & appropriateness | ► Perception of implementation stakeholders (participants and non-participants) that the CRC-SIM is agreeable and satisfactory<br>► Perceived fit/ relevance or compatibility of the CRC-SIM for clinic and the target population | ► Interview<br>► Analysis of participant information and implementation outcomes |
| Feasibility | ► Extend to which the CRC-SIM can be successfully used within the community and clinics<br>► Demand on provider/system<br>► Administrative data | ► Monitoring and recording of data by research staff |

Continued

| Outcome measure | Details | Mode of assessment |
|---|---|---|
| Budget-impact analysis | <u>Current screening scenario (Ministry of Health Malaysia)</u><br>▶ No of persons assessing the patients<br>▶ Time to make an assessment (assessing symptoms, taking family history, recommending iFOBT, analysing iFOBT referring for colonoscopy, assessment for colonoscopy, conducting colonoscopy)<br>▶ Gross income of the persons conducting the testing/ screening<br>▶ Cost of iFOBT/ colonoscopy/biopsy<br>▶ Cost of disposal of tests<br>▶ Number of iFOBTs/ colonoscopies conducted<br><u>Intervention scenario (CRC-SIM)</u><br>▶ Number and duration of reminder calls and texts<br>▶ Time to contact participants (recruitment and follow-up)<br>▶ Cost of the screening pack<br>▶ Cost of distributing screening pack<br>▶ Time to feedback results<br>▶ Cost of iFOBT/colonoscopy/ biopsy<br>▶ n and type of staff involved to deliver the programme<br>▶ Gross income of staff involved in programme (nurse/ research assistant/medical officers)<br>▶ Number of iFOBTs/colonoscopies conducted<br>▶ Cost to design and print study materials<br>▶ Cost of disposal of kit | ▶ Survey of healthcare facilities and laboratory<br>▶ Data recorded by research staff |

CRC-SIM, colorectal cancer screening intervention for Malaysia; FOBT, feacal occult blood test; iFOBT, immunochemical feacal occult blood test; SEACO, South East Asia Community Observatory.

that are received by the team. This procedure will be repeated for every 25th photo. In the event that agreement is not reached between the two researchers, a laboratory staff member who is experienced in interpreting iFOBT results will be consulted until 100% agreement is reached. Follow-up clinic and colonoscopy attendance by iFOBT positive participants will be collected from KK Segamat and Hospital Segamat (table 1). Participants with iFOBT positive results will also be contacted over the phone by trained SEACO research staff 1 month after the referral to confirm colonoscopy attendance.

*Budget impact analysis*

A health service perspective will be adopted for an incremental budgetary impact analysis undertaken in a fashion consistent with best practice guidance. Each element of resource used will be itemised as will the unit costs applied to it. Costs will be examined over a 5-year time frame and will focus on those directly attributable to the delivery of screening that fall on the health service (table 1). The target population will be those aged 50–75; activities to be compared will cover recruitment, screening, communication of results and disposal of test materials.

In brief, the numbers and grades of staff employed to provide current operations (including overheads) will be obtained from records and surveys of the participating study site. Staff time will be recorded and valued based on the salaries (including overheads) of those involved

in screening and reflect both the time involved in the conduct of tests and the communication of results. To this will be added the cost of consumables—test packs, gloves, masks, etc together with costs associated with the disposal of materials from the test. These will be estimated on a per-individual screened basis, with the number of individuals screened in a year also collected (see table 1). For the intervention, the same elements of cost will be gathered via survey of participating sites. To these will be added the cost of educational and recruitment materials based on study records and estimated for delivery at scale, that is, reflective of the potential for economies of scale in the production of printed materials. Staff time in assessing the test result will again be based on the salary and time taken by those involved, which will be calculated based on a time and motion study. For 'usual care', staff time spent on recruitment is likely to be minimal given that a doctor will simply ask a patient if they would like to undertake a test. Similarly, there do not appear to be materials related to recruitment via the usual opportunistic option and these will be assumed to be zero.

In sensitivity analyses, staff time involved in delivery of screening will be varied as will unit costs applied to staff time, the cost of iFOBT, colonoscopies and biopsies. The cost of calls will be based on usual service charges as will text messages. In sensitivity analysis a zero cost will be applied to those communications undertaken via

WhatsApp usage. The target population will be assumed to be those aged 50–75 and will be estimated from census records. Uptake will be based on that observed within the study scaled up for the population to generate estimates of total costs under the two scenarios over 5 years as well as the incremental difference in these.

## Qualitative analysis

Interviews with participants and key stakeholders will be analysed thematically in NVivo V.12. Two researchers will code transcripts independently, early coding will be shared and discussed iteratively until agreement is reached; the coding framework will be applied to the remaining transcripts; and themes will be discussed with the research team in order to map out key themes and subthemes that represent the interview data and views of stakeholders and participants.

## Statistical analysis

Quantitative data will be analysed with SPSS V.24. Descriptive statistics at baseline will be reported as mean (SD) for continuous data and frequencies (percentages) for categorical data. The true percentage uptake will be determined along with 95% Confidence Intervals (CIs). Logistic regression models will be used to investigate variables, which influence (eg, participant characteristics) screening completion. Ideally, we would like to select randomly one person from each household to avoid clustering but available resources and the rural setting led us to the pragmatic decision to sample every eligible person in each sampled household. We will control for potential within household clustering. CIs will be calculated using variance inflation factors to account for the lack of independence of each person in the same household (using the svy function in STATA).[27]

## Sampling size and procedure

It is estimated that a sample size of approximately 780 participants will allow the true percentage uptake of screening to be determined with 95% CI within plus or minus 2.1% (ie, the 95% CI will span 4.2 percentage points if the uptake is around 90%); and provide a precise estimate of the true percentage uptake of screening by income and ethnic group, for example, in the poorest 40% (based on median monthly income 3000RM/£600), uptake of screening will be determined within ±6%, and, in the Chinese, uptake of screening will be determined within ±7%. Random sampling of households will be applied to select the study population from the SEACO database.

## Data quality assurance and control

Each participant will be assigned a study ID and personal data will not be stored or shared together with study outcomes (ie, iFOBT photographs and colonoscopy findings). The names of participants will be stored separately from any study information in order to safeguard anonymity and confidentiality. One SEACO staff member will be in charge of monitoring the confidential SEACO phone number to which participants send iFOBT results. Photographs of the results will be stored on a google drive and access will be restricted to project research staff only. The data collector supervisor will randomly select recorded recruitment calls for the purpose of quality assurance and discuss any improvement suggestions with data collectors.

## DISCUSSION

The CRC-SIM is a collaborative research effort between academics, public and private healthcare, and the MOH Malaysia. It is the first study to test the implementation of a CRC screening intervention in Malaysia in the context of the COVID-19 pandemic. The combination of a digital intervention combined with CRC home-testing, is innovative and, to our knowledge, unique in LMICs. Postal CRC screening interventions have been tested primarily in high-income countries (HICs) and have been proven to be effective in a number of upper-middle-income countries. Malaysia and Thailand are the only South East Asian countries that have published CRC screening studies and used a face-to-face recruitment approach[18] except for one study that tested recruitment through pamphlets that were distributed in public places.[28] Public Health CRC screening interventions in HICs have been associated with a significantly greater screening uptake compared with controls (OR 1.49, 95% CI 1.43 to 1.56) and interventions providing instructions or demonstration of the targeted behaviour and that provided objects (eg, letter, leaflet, screening kit) produced stronger effects.[29] Interventions that utilised telephone contact, doctors' endorsement and simplified test procedures were effective intervention strategies for increasing participation in mail-out CRC screening interventions.[30] LMICs have implemented mostly opportunistic screening due to a lack of healthcare resources in the public healthcare system, including testing facilities and trained clinical staff. Self-testing has been introduced for pregnancy tests, HIV tests[31 32] and the management of NCDs such as diabetes and hypertension. Due to the COVID-19 pandemic, self-testing is gaining further attention and has been introduced, globally, to combat the spread of the virus.[33 34] Self-testing for CRC, guided by health professionals, is likely to reduce pressure on the healthcare system (eg, doctor–patient time and laboratory staff time to analyse stool samples) as well as outlay for individual patients (eg, transport cost and travel time to clinic) in Malaysia. The findings from this study will be used to contribute a blueprint for the implementation of home-testing interventions for CRC screening in LMICs. The results of the BIA may be used by policy makers and service planners to inform discussion about the affordability of this mode of screening though it is important to note that a further evaluation would be required at some stage to assess its cost-effectiveness and value for money.

It is important to note that the conduct of the evidence synthesis that informed the design of the intervention

began shortly before the MCO was introduced in Malaysia as a public health measure to control the COVID-19 outbreak. Although the COVID-19 pandemic has limited the opportunity for face-to-face contact that patients prefer, it is providing opportunities to test digital health interventions that rely on the use of small and mass media.[35] In 2019, 98.2% of the population had access to a mobile phone and 91% of those had access to a smartphone and 87% access to mobile broadband.[36] Due to the restriction of movement in Malaysia, it is likely that smartphone ownership and access to mobile broadband will increase further and facilitate participation and coverage due to the use of contact tracking apps recommended by the government during the pandemic. Since the main aim of this study is to test the implementation of the intervention, outcome measures such as iFOBT completion and colonoscopy attendance will be used to report on the appropriateness and feasibility of the intervention. WhatsApp is used increasingly for research purposes in LMICs, mainly to share links to online surveys[37] and it is the most commonly used platform for personal contact and exchange in Malaysia. We will communicate via text message to participants who do not have WhatsApp. The SEACO database is a major advantage in the evaluation of the intervention. For example, it provides population level data and a frame for random sampling.

There is an urgent need to improve CRC screening in South East Asia and to address disparities and preferences particularly in ethnically diverse countries like Malaysia.[6 38] Creating an intervention video in the three major local languages is likely to be more culturally acceptable and address communication barriers that would be difficult to address in a clinical setting.

### Ethics and dissemination

Ethics approval has been granted by the Monash University Human Research Ethics Committee (MUHREC, ID: 29107) and the Medical Review and Ethics Committee (MREC ID: 21-02045-O7G(2)). Consent to participate will be gathered verbally at the time of recruitment due to telephone recruitment. Gathering verbal consent has been approved by MUHREC. Participants will be assigned unique identification numbers to ensure anonymity and confidentiality of data collected. All participants will be informed that they may withdraw from the study any time without giving any reason.

We will disseminate the findings from the CRC-SIM at meetings with the MOH, local clinics and communities in Segamat as well as in the SEACO community newsletter and peer-reviewed journals and conferences

**Author affiliations**
[1]Centre for Public Health and UKCRC Centre of Excellence for Public Health, Queen's University Belfast, Belfast, UK
[2]Global Public Health, Jeffrey Cheah School of Medicine and Health Sciences, Monash University Malaysia, Subang Jaya, Malaysia
[3]South East Asia Community Observatory (SEACO), Jeffrey Cheah School of Medicine and Health Sciences, Monash University Malaysia, Subang Jaya, Malaysia
[4]Cancer Unit, Disease Control Division, Ministry of Health Malaysia, Putrajaya, Malaysia
[5]Surgical Department, Hospital Segamat, Segamat, Malaysia
[6]District Health Office (PKD), Segamat, Johor, Malaysia

**Acknowledgements** Thanks to Dr Tracy Owen, Consultant in Public Health Medicine and lead for the NI Bowel Cancer Screening Programme in the NI Public Health Agency and to all key stakeholders and community members for their input.

**Contributors** MD and TTS conceptualised the project. TTS, KR, DS, MD and NSBIT planned, coordinated and conducted the study. PN and ANbMG contributed to the discussion about the design and implementation of the intervention and its evaluation. DS drafted the manuscript. CO'N planned the cost analysis and CRC and MD planned the statistical analysis and calculated the sample size. RI provided information regarding the SEACO database and conducted the sampling. MD, TTS, NSBIT, CRC, CO'N, FK and KR contributed to the protocol design and reviewed and edited the manuscript. All authors read and approved the final manuscript.

**Funding** This research was supported by the Medical Research Council (UK) Global Challenges Research Fund (MR/S014349/1). The collaborative grant application was subjected to peer-review by individual academic reviewers and the final decision about funding was made by an expert panel.

**Disclaimer** The funder had no role in the design of the study, collection, analysis, and interpretation of data or in writing the manuscript.

**Competing interests** None declared.

**Patient and public involvement** Patients and/or the public were involved in the design, or conduct, or reporting, or dissemination plans of this research. Refer to the Methods section for further details.

**Patient consent for publication** Not applicable.

**Provenance and peer review** Not commissioned; externally peer reviewed.

**ORCID iDs**
Désirée Schliemann http://orcid.org/0000-0002-8746-3002
Tin Tin Su http://orcid.org/0000-0003-0387-6406

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
