## [Reviewer comments · BMJ Open]

ARTICLE DETAILS

TITLE (PROVISIONAL)	Implementation of a colorectal cancer screening intervention in Malaysia (CRC-SIM) in the context of a pandemic: study protocol
AUTHORS	Schliemann, Desiree; Ramanathan, Kogila; Ibrahim Tamin, Nor Saleha; O'Neill, Ciaran; Cardwell, Christopher; Ismail, Roshidi; Nanu, Prathibha; bin Mohd Ghazali, Ahmad Nizal; Kee, Frank; Su, Tin; Donnelly, Michael

VERSION 1 – REVIEW

REVIEWER	Roselle De Guzman Manila Central University-Filemon D Tanchoco Medical Foundation Hospital
REVIEW RETURNED	03-Dec-2021

GENERAL COMMENTS	Thank you for the opportunity to review the manuscript. The research addresses an unmet need on CRC screening in low-middle-income countries particularly in the pandemic situation. The study is significant and innovative. It focuses on the population-based cancer screening. It also addresses the relevant socio-cultural barriers to CRC screening in the local community. The research strategy and details of methodology are scientifically sound. The approach is transferable and adoptable to other resource-limited settings. This is a well- and clearly written manuscript. Current and relevant research studies and references are cited. Methods are meticulous and sufficiently detailed in separate sections. The results are clearly presented as well. I would prefer to know more about the ethics oversight for the study. It would be more interesting to have some discussion on study the strengths and limitations. I support the publication of this study.
--

REVIEWER	Ajay Kumar Khanna Banaras Hindu University
REVIEW RETURNED	14-Apr-2022

GENERAL COMMENTS	Comments on the manuscript The manuscript title: Implementation of a colorectal cancer screening intervention in Malaysia (CRC-SIM) in the context of a pandemic: study protocol 1. The study protocol submitted for review by the authors aims to conduct implementation research for improving colorectal cancer (CRC) screening uptake in Malaysia. They conducted an evidence synthesis and local needs assessment and will conduct a screening
--

	intervention on a sample of 780 randomly selected participants from the SEACO database, aged 50-75 years and living in Segamat district, Malaysia. 2. The study model is well-planned and is in line with the RE-AIM framework and the description of intervention has been as per the TIDieR checklist. 3. The screening intervention utilises a home-testing kit and digital communication through Whatsapp to improve CRC screening uptake. 4. Roles of study team: The roles of SEACO staff and research assistants as intervention providers will be to recruit participants over the phone based on eligibility, send a text message reminder after one week to all participants to complete the test and share the result and those who did not share their iFOBT result after two weeks will be called and reminded by the research assistant. The results will be shared with the participants through text messages if negative and telephone calls if positive. Participants with a positive result will receive a text message containing a photograph of a referral letter. The staff will conduct interviews with both study participants and health care staff post-intervention as a part of follow-up. Two trained medical professionals who are members of the research team will interpret the results independently and compare them for accuracy. Those participants who screen positive will visit their doctor at a public health clinic who will likely refer the participants to the hospital colonoscopy service Following are the specific comments: 1. It would be good if the investigators can briefly describe the following for the intervention providers (as per the ITEM-5 of the TIDieR checklist) : 2. The number of providers involved in delivering or undertaking the intervention; 3. Their disciplinary background or what pre-existing specific skills, expertise, and experience providers required. 4. How they were trained for the intervention and how their competence was assessed and if found lacking what methodology they adopted for improving competency. Will there be any training workshops for them? 5. If the SEACO staff will be doing the intervention as part of their normal role or were specially recruited as providers for purposes of the study; whether providers were reimbursed for their time or provided with other incentives (if so, what) to deliver the intervention as part of the study, and whether such time or incentives might be needed to replicate the intervention. 6. Is there any mechanism for those participants whose tests results come out invalid/inconclusive? 7. Eligibility criteria: Should the participants with previous colonic examination and recent (less than one year) FiOBT test, those with a cognitive and physical disability to perform the test be considered as exclusion. 8. Will there be any risk categorization for the participants? Previous studies have reflected that in countries with low-resource settings identification of high-risk populations and the development of risk-adapted screening strategies may be more cost-efficient than conventional screening strategies. Also, risk categorization is already in the CRC screening pathway of Malaysia as mentioned in figure-1. 9. What will be the mechanism for the participants who wish to refuse/withdraw consent for testing?
--	--

	10. How and when will the participants be informed about the Research Question, study methods and screening intervention and the cost that will be borne by them if any? 11. Was the decision to go to the local health clinic once the test comes positive will be based on the clinic nominated by the participant? If not what mechanism decides that screen positive participants will visit which clinic for an initial assessment. 12. Since the FiOBT test is required to be done annually, will there be follow up after study completion to assess/monitor the maintenance of the screening intervention by the community and health care staff? 13. Timelines concerning study phase/status (what all has been done so far, when will the recruitment start, if started how many recruited so far, expected duration of the study, follow-updates) need to be mentioned. 14. For quality assurance and control what all measures will be adopted? Especially in regards to data management (data confidentiality and safety) and the involvement of experts from epidemiology, endoscopy, pathology, surgery and social science. 15. The study seems like a single-arm intervention study hence there is a potential limitation of no outcome comparison between groups. If the study setting has some conventional screening rates/ data can it be used for outcome comparison? 16. How and when will the satisfaction or attitude of the participants in regards to this screening intervention be assessed? 17. Whatsapp usage has a very good penetration in Malaysia as seen in previous reports. It has a good potential to be a health care provider and communication tool, however, there are certain limitations related to patient data safety and privacy. Investigators should ensure that these areas are duly considered. 18. Though the sample size is small to generate cancer incidence rates investigators should mention the treatment pathways for those participants who are diagnosed with colon during the study.
--	--

REVIEWER	Nawi Ng University of Gothenburg
REVIEW RETURNED	14-Apr-2022

GENERAL COMMENTS	REVIEW FOR BMJ OPEN L124-L127: As the Indian population is underrepresented in the study area (2%), shouldn't they be oversampled in the intervention so that the researchers could assess the intervention's acceptability and feasibility. Outreach for screening to the Indian population might differ from other populations, and they might have different attitudes towards screening. L132-L133: It is unclear how the researchers could assess individuals' experience with CRC symptoms in ascertaining their exclusion criteria. What symptoms are included? How will they be measured? L137: The contents of the section "Conceptual Framework" are not related to any conceptual frameworks that are relevant to the study. The authors could update the heading of the section, and a section on "Conceptual Framework" should be included. L144-L151: The rationale for reviewing the CRC screening programme in Northern Ireland in this study protocol should be provided. How could the review be relevant to the study in a different
--

	context in Malaysia? What lessons were learnt from the review in Ireland that could be used in the Malaysian context? L181-196: The authors could have presented the qualitative findings for men and women with different ethnic backgrounds. I believe the findings from different ethnic groups are different and of utmost importance to guide tailoring the intervention designs. For example, trust in health care organisations and financial concerns related to cancer screening and treatment are different among the different ethnic groups in Malaysia. L197-L211: Please be more balanced in presenting the results from the interviews with key stakeholders. The results presented in the manuscript are barriers to CRC screening. Didn't the interviewees also provide their perception of the enabling factors? L231-L232: I wonder about the value of the random selection of the participants. Wouldn't a stratified random sampling be better to ensure the representation of all population groups, especially the Indian population? See related comment above. L241-L243: Please clarify what "approach" was referred to on Line 241. And how does the "approach" address inhibiting factors and utilise facilitating factors? It is unclear what the authors meant in that section. L267-L268: Please provide more details on the design of the mixed-method approach used. Will the authors design a sequential or a concurrent mixed method design? If sequential, whether it will be exploratory or explanatory? The rationale should match the design presented subsequently. L275-L276: Please clarify the sentence "including group comparison of residents who agreed or refused". Did the author plan to have a comparison or control group in the intervention? If so, this should be illustrated in the Figure on Page 29. L279-L292: The choice to conduct interviews to assess the uptake of the intervention only among a sub-sample of the study participants should be motivated. Shouldn't the researcher conduct the interviews also among the participants who comply with all intervention strategies? It is unclear how the health care staff could be recruited through email (L286). L295-L296: Please provide more details on how the two medical professionals resolve the disagreement "until 100% agreement is reached"? Will a third reviewer be brought in to solve the disagreement? L353-L355: What advanced statistical techniques are to be used? Please state this clearly in the paper. L359-L365: Please clarify the sample size calculation. I assume the authors used the sample size formula for one proportion. Based on the authors' assumption, they should end up with a minimum sample size of 865 individuals ($=0.9*0.1*(1.96/0.02)^2$). They might also need to account for the response rate and might have a larger sample size. L369-L370: The paper claims that CRC-SIM is the first study to test
--	---

	the implementation of a CRC screening in Malaysia in the context of the COVID-19 pandemic. The pandemic also appears in the title but has been mentioned in passing throughout the paper. Please specify how the pandemic has influenced the design and implementation of the intervention in the paper. L397-L398: How could smartphone ownership and access to mobile broadband be increased due to the restriction of movement (assumed due to Covid-19)? Table 1 and the intervention: The research has planned to use SEACO staff to implement the intervention. This design poses a threat to the external validity of the findings. One could assume that SEACO staff have different qualifications than the health staff working in primary care or community care. Could the results of CRC-SIM be scalable to the broader health care setting if adoption and implementation of the intervention are assessed only among SEACO staff but not among staff working under the MoH who are going to implement this CRC-screening intervention if it is scaled up? There are parts of the texts which fit better elsewhere. For example: - L105-107 on the qualitative interview: Shouldn't this fit better in the "Methods of assessment" section? - L350 on statistical analysis: Will the qualitative data be analysed statistically in NVivo? This sentence does not fit in this section.
--	--

VERSION 1 – AUTHOR RESPONSE

Reviewer: 1

Dr. Roselle De Guzman, Manila Central University-Filemon D Tanchoco Medical Foundation Hospital
Comments to the Author:

Thank you for the opportunity to review the manuscript. The research addresses an unmet need on CRC screening in low- middle-income countries particularly in the pandemic situation. The study is significant and innovative. It focuses on the population-based cancer screening. It also addresses the relevant socio-cultural barriers to CRC screening in the local community. The research strategy and details of methodology are scientifically sound. The approach is transferable and adoptable to other resource-limited settings.

This is a well- and clearly written manuscript. Current and relevant research studies and references are cited. Methods are meticulous and sufficiently detailed in separate sections. The results are clearly presented as well.

I would prefer to know more about the ethics oversight for the study. It would be more interesting to have some discussion on study the strengths and limitations.

I support the publication of this study.

Reply: Thanks a lot for your feedback. We have added the reference number from the ethical approval (under ethics and dissemination). Strengths and limitations of the study are listed on page 3.

Reviewer: 2

Dr. Ajay Kumar Khanna, Banaras Hindu University
Comments to the Author:
Dear Editor

Following are the specific comments:

1. It would be good if the investigators can briefly describe the following for the intervention providers (as per the ITEM-5 of the TIDieR checklist) :
2. The number of providers involved in delivering or undertaking the intervention;

Reply: We included this information under 'Training and piloting'.

3. Their disciplinary background or what pre-existing specific skills, expertise, and experience providers required.

Reply: We included this information under 'Training and piloting'.

4. How they were trained for the intervention and how their competence was assessed and if found lacking what methodology they adopted for improving competency. Will there be any training workshops for them?

Reply: We included this information under 'Training and piloting'.

5. If the SEACO staff will be doing the intervention as part of their normal role or were specially recruited as providers for purposes of the study; whether providers were reimbursed for their time or provided with other incentives (if so, what) to deliver the intervention as part of the study, and whether such time or incentives might be needed to replicate the intervention.

Reply: SEACO staff received a full-time salary. The salary of staff and any other costs involved in the study will be published separately as part of a budget impact analysis.

6. Is there any mechanism for those participants whose tests results come out invalid/inconclusive?

Reply: We have addressed this in the protocol.

7. Eligibility criteria: Should the participants with previous colonic examination and recent (less than one year) iFOBT test, those with a cognitive and physical disability to perform the test be considered as exclusion.

Reply: We included all exclusion information under the participant information. We applied current opportunistic screening criteria to the study population.

8. Will there be any risk categorization for the participants? Previous studies have reflected that in countries with low-resource settings identification of high-risk populations and the development of risk-adapted screening strategies may be more cost-efficient than conventional screening strategies. Also, risk categorization is already in the CRC screening pathway of Malaysia as mentioned in figure-1.

Reply: We will follow the current screening guidelines in Malaysia regarding the implementation of the CRC-SIM, i.e. participants who describe CRC symptoms or have a family history of CRC will be excluded from the study. Participants with symptoms will be advised to speak to their doctor (who according to the screening guidelines will recommend people to have a colonoscopy). We haven't added any further risk stratification to this study in order to reflect as close as possible the current screening scenario including qualifying iFOBT screening guidelines.

9. What will be the mechanism for the participants who wish to refuse/withdraw consent for testing?

Reply: We added relevant information to the protocol under 'participant information'

10. How and when will the participants be informed about the Research Question, study methods and screening intervention and the cost that will be borne by them if any?

Reply: We added information to the protocol methods under 'Intervention'.

11. Was the decision to go to the local health clinic once the test comes positive will be based on the clinic nominated by the participant? If not what mechanism decides that screen positive participants will visit which clinic for an initial assessment.

Reply: The chosen health clinic was the one that was closest to both study districts and therefore closest to participants. We specified this under 'intervention'.

12. Since the FiOBt test is required to be done annually, will there be follow up after study completion to assess/monitor the maintenance of the screening intervention by the community and health care staff?

Reply: Unfortunately, there are no resources at this stage to monitor future screening of participants. Due to confidentiality reasons, we are also not able to share with the clinic who has completed the screening test unless the participant has been informed that they tested iFOBt positive. All participants will be encouraged to go to the clinic annually for continued screening. We included that information under 'intervention'.

13. Timelines concerning study phase/status (what all has been done so far, when will the recruitment start, if started how many recruited so far, expected duration of the study, follow-up updates) need to be mentioned.

Reply: We submitted this protocol to BMJ Open in October 2021 at which stage the intervention was still ongoing. We now have completed all data collection and conclude the intervention.

14. For quality assurance and control what all measures will be adopted? Especially in regards to data management (data confidentiality and safety) and the involvement of experts from epidemiology, endoscopy, pathology, surgery and social science.

Reply: We added a section on 'data quality assurance and control' to address this comment.

15. The study seems like a single-arm intervention study hence there is a potential limitation of no outcome comparison between groups. If the study setting has some conventional screening rates/ data can it be used for outcome comparison?

Reply: Local screening data has not been published and we are unable to include this information.

16. How and when will the satisfaction or attitude of the participants in regards to this screening intervention be assessed?

Reply: As mentioned under 'interviews', a selected sub-sample of participants will be interviewed after intervention completion.

17. Whatsapp usage has a very good penetration in Malaysia as seen in previous reports. It has a good potential to be a health care provider and communication tool, however, there are certain limitations related to patient data safety and privacy. Investigators should ensure that these areas are duly considered.

Reply: We communicated to participants details about the nature and content of the intervention including the use of WhatsApp and we informed participants that Information that can be used to identify individuals will not be shared outside of the research team. All information will be kept with strict confidentiality and will only be used for the research purpose. No information that could lead to the identification of you will be disclosed in any published reports or findings.

18. Though the sample size is small to generate cancer incidence rates investigators should mention the treatment pathways for those participants who are diagnosed with colon during the study.

Reply: We included this information under 'intervention'.

Reviewer: 3

Dr. Nawi Ng, University of Gothenburg

Comments to the Author:

REVIEW FOR BMJ OPEN

L124-L127: As the Indian population is underrepresented in the study area (2%), shouldn't they be oversampled in the intervention so that the researchers could assess the intervention's acceptability and feasibility. Outreach for screening to the Indian population might differ from other populations, and they might have different attitudes towards screening.

Reply: We clarified this under 'study population'

L132-L133: It is unclear how the researchers could assess individuals' experience with CRC symptoms in ascertaining their exclusion criteria. What symptoms are included? How will they be measured?

Reply: This information was self-reported by the participant. We included further information under 'study population'

L137: The contents of the section "Conceptual Framework" are not related to any conceptual frameworks that are relevant to the study. The authors could update the heading of the section.

Reply: We have amended the headings accordingly.

L144-L151: The rationale for reviewing the CRC screening programme in Northern Ireland in this study protocol should be provided. How could the review be relevant to the study in a different context in Malaysia? What lessons were learnt from the review in Ireland that could be used in the Malaysian context?

Reply: We learned that there is not a direct read across from NI and the wider UK context to the context in Malaysia though the comparison was instructive about, for example, the screening materials and operational procedures that are used in the NI and wider UK context and how aspects of these processes and materials could be adapted for use in Malaysia. In addition, we learned about potential inhibiting and facilitative factors and their evaluation.

L181-196: The authors could have presented the qualitative findings for men and women with different ethnic backgrounds. I believe the findings from different ethnic groups are different and of utmost importance to guide tailoring the intervention designs. For example, trust in health care organisations and financial concerns related to cancer screening and treatment are different among the different ethnic groups in Malaysia.

Reply: The findings from the qualitative research will be published separately and the results will be separately discussed for men, women and different ethnic groups where appropriate.

L197-L211: Please be more balanced in presenting the results from the interviews with key stakeholders. The results presented in the manuscript are barriers to CRC screening. Didn't the interviewees also provide their perception of the enabling factors?

Reply: Yes, the enabling factors are presented at the end of the paragraph. We are preparing a publication of the interviews so that if the reader is interested in more detail, they will be able to access that information.

L231-L232: I wonder about the value of the random selection of the participants. Wouldn't a stratified random sampling be better to ensure the representation of all population groups, especially the Indian population? See related comment above.

Reply: The sample that was drawn from the SEACO database by our statistician was representative of the main ethnic groups and he advised that when stratified random sampling was applied, a similarly representative population profile was obtained.

L241-L243: Please clarify what "approach" was referred to on Line 241. And how does the "approach" address inhibiting factors and utilise facilitating factors? It is unclear what the authors meant in that section.

Reply: Thanks you for pointing that out. We referred to the intervention and have updated this in the protocol.

L267-L268: Please provide more details on the design of the mixed-method approach used. Will the authors design a sequential or a concurrent mixed method design? If sequential, whether it will be exploratory or explanatory? The rationale should match the design presented subsequently.

Reply: This is a concurrent approach and we have included that information in the qualitative section.

L275-L276: Please clarify the sentence "including group comparison of residents who agreed or refused". Did the author plan to have a comparison or control group in the intervention? If so, this should be illustrated in the Figure on Page 29.

Reply: No, there is no control group but in the analysis we planned to compare people who agreed to participate in the study vs those that refused and also participants who completed the iFOBT and those that did not.

L279-L292: The choice to conduct interviews to assess the uptake of the intervention only among a sub-sample of the study participants should be motivated. Shouldn't the researcher conduct the interviews also among the participants who comply with all intervention strategies? It is unclear how the health care staff could be recruited through email (L286).

Reply: That's correct, we aimed to conduct interviews will a sub-sample of participants that completed/ did not complete all components of the intervention and have clarified this in this section.

L295-L296: Please provide more details on how the two medical professionals resolve the disagreement "until 100% agreement is reached"? Will a third reviewer be brought in to solve the disagreement?

Reply: We have clarified this section.

L353-L355: What advanced statistical techniques are to be used? Please state this clearly in the paper.

Reply: We have revised this section.

L359-L365: Please clarify the sample size calculation. I assume the authors used the sample size formula for one proportion. Based on the authors' assumption, they should end up with a minimum sample size of 865 individuals ($=0.9 \cdot 0.1 \cdot (1.96/0.02)^2$). They might also need to account for the response rate and might have a larger sample size.

Reply: We have revised this section.

L369-L370: The paper claims that CRC-SIM is the first study to test the implementation of a CRC screening in Malaysia in the context of the COVID-19 pandemic. The pandemic also appears in the title but has been mentioned in passing throughout the paper. Please specify how the pandemic has influenced the design and implementation of the intervention in the paper.

Reply: We added further information in the intervention section to highlight how the pandemic informed the study design.

L397-L398: How could smartphone ownership and access to mobile broadband be increased due to the restriction of movement (assumed due to Covid-19)?

Reply: We clarified this part in the discussion.

Table 1 and the intervention: The research has planned to use SEACO staff to implement the intervention. This design poses a threat to the external validity of the findings. One could assume that SEACO staff have different qualifications than the health staff working in primary care or community care. Could the results of CRC-SIM be scalable to the broader health care setting if adoption and implementation of the intervention are assessed only among SEACO staff but not among staff working under the MoH who are going to implement this CRC-screening intervention if it is scaled up?

Reply: This is an important point that we recognised at the outset. However, in the context of the available resources and discussions with the Ministry of Health representative on our steering group, the use of SEACO staff was the only way in which we could test in principle, a mode of CRC home-testing and its feasibility and acceptability, particularly during the pandemic. It seems reasonable to assume that participation in home testing by clinic staff will be at least comparable to SEACO staff. In addition, we plan to include the cost of using clinic staff instead of SEACO staff in our BIA.

There are parts of the texts which fit better elsewhere. For example:

- L105-107 on the qualitative interview: Shouldn't this fit better in the "Methods of assessment" section?

Reply: We have moved this as suggested.

- L350 on statistical analysis: Will the qualitative data be analysed statistically in NVivo? This sentence does not fit in this section.

Reply: We have removed this here.